# Why and How Should We Assess the Cardiovascular Risk in Patients with Juvenile Idiopathic Arthritis? A Single-Centre Experience with Carotid Intima-Media Measurements

**DOI:** 10.3390/children10030422

**Published:** 2023-02-22

**Authors:** Marta Gruca, Krzysztof Orczyk, Justyna Zamojska, Katarzyna Niewiadomska-Jarosik, Jerzy Stańczyk, Elżbieta Smolewska

**Affiliations:** 1Department of Pediatric Cardiology and Rheumatology, Medical University of Lodz, 91-738 Łódź, Poland; 2Department of Pediatric Infectious Diseases, Medical University of Lodz, 91-347 Łódź, Poland

**Keywords:** carotid intima-media thickness, juvenile idiopathic arthritis, cardiovascular risk

## Abstract

Background: Children diagnosed with juvenile idiopathic arthritis (JIA) are thought to be more likely to develop cardiovascular disease in adulthood. The factors modulating the cardiovascular risk, involving exposure to secondhand smoking, sedentary lifestyle and abnormal body mass index, might have had a stronger impact during the COVID-19 pandemic. The lack of reliable prognostic markers for a higher probability of cardiovascular events might be solved by carotid intima-media thickness (cIMT) measurement. The paramount goal of the study was to assess its usefulness in JIA patients. Materials and Methods: The results of cIMT measured by a single physician in 45 children diagnosed with JIA were compared to 37 age- and sex-matched healthy counterparts. The analysis also involved anthropometric parameters, laboratory tests, and a survey regarding lifestyle-related factors. Results: Four JIA patients appeared to have cIMT above the 94th percentile. A positive correlation between erythrocytes sedimentation rate (ESR) and right carotid artery percentiles was found. Passive smoking increased the cardiovascular risk regardless of JIA. Doubling the daily screen time during the pandemic led to a significant reduction in children’s physical activity. However, the number of enrolled subjects was not enough to make significant recommendations. Conclusions: cIMT measurements remain an interesting perspective for future cardiovascular screening of children with JIA. It has yet to be determined whether it should be considered in all JIA patients on a reliable basis.

## 1. Background

Despite the undeniable progress in therapeutic possibilities for patients diagnosed with the most common arthropathy in childhood, juvenile idiopathic arthritis (JIA), a substantial proportion of patients develop mild to moderate long-term disability with a decreased health-related quality of life [1]. Similarly to other autoimmune juvenile-onset illnesses, including type 1 diabetes and systemic lupus erythematosus, JIA patients are deemed to be at higher risk of developing cardiovascular disease (CVD) during the adulthood [2]. As for adults with rheumatoid arthritis (RA), there was a 48% increased risk of incident CVD involving acute myocardial infarction, cerebrovascular events, and congestive cardiac failure [3]. They were also more likely to develop long QT syndrome when compared to healthy controls [4]. Cardiovascular involvement in JIA may affect all parts of the heart, including peri-, myo- and endocardium, valves, and coronary arteries [5]. The cardiac conduction system may as well be engaged; however, the current findings regarding QT intervention in children with JIA remain ambiguous [6].

Early recognition of subclinical abnormalities in cardiac function and structure is essential for the prevention of apparent cardiovascular symptoms and worse long-term health outcomes [7]. Active inflammation within JIA is combined with the overproduction of proinflammatory cytokines, involving tumor necrosis factor and interleukins 1 and 6, which may subsequently promote endothelial dysfunction, playing a crucial role in atherogenesis [8]. Importantly, the levels of proinflammatory molecules may remain elevated even during the remission of arthritis [9]. Duffy et al. noted that arthritis is very likely to persist in adulthood if the remission does not occur within 10 years of disease onset, which can be achievable only in approximately 30% to 35% of patients [10]. Furthermore, young adults with RA have been reported to develop subclinical signs of atherosclerosis [11]. Given that persistent inflammation combined with negative environmental factors may result in a higher incidence of cardiovascular accidents [12], the promotion of ideal health behavior in JIA patients seems to be underrated. Cardiovascular prevention should be defined as a coordinated set of actions designed for the population or selected individuals to eliminate or minimize the consequences of and disabilities caused by CVD [13]. In 2015, the costs of managing CVD and their sequelae were estimated to be 210 billion euros in the whole European Union [14]. It consumed 16% of healthcare expenditures in Poland, compared to 8% in Germany, 7% in France, and only 3% in Denmark and Sweden [14]. The two main compounds of these costs are hospitalizations and pharmacotherapy [15]. An efficient system of prevention would reduce the treatment expenses related to CVD [16].

The principal objective of CVD prophylaxis in the pediatric population is to maintain an ideal state of cardiovascular health [17]. In 2016, The American Heart Association proposed a set of metrics in children and adolescents to define cardiovascular health [17]. It included: Abstinence from smoking, a normal body mass index (BMI), regular physical activity levels, a proper diet (containing fruits, vegetables, fish, whole grains, and low sodium and low sugar foods and drinks), a low concentration of total cholesterol, proper blood pressure (BP), and a normal fasting glucose level. The assessment of the listed components during the routine visit of a JIA patient in rheumatology outpatient clinic might be a useful screening method for differentiating children with a substantially higher cardiovascular risk [18]. However, meticulous history taking may not be sufficient to evaluate the full impact of the COVID-19 pandemic on the patients’ behavior. The temporary closure of schools and sports facilities along with the sudden transition to online learning must have affected the frequency of physical activity as well as dietary habits and screen exposure time [19,20]. These alterations might have influenced previous cardiovascular risks in JIA patients. Therefore, there is a need to determine a new, easily-accessible tool which can be utilized to identify children who require closer cardiology care.

Carotid intima-media thickness (cIMT) measurement is one of the best verified non-invasive methods of prognosing atherogenesis and evaluating cardiovascular risk [21]. The strong correlation between the dimensions obtained with ultrasonography and the vessel thickness in histological samples, followed by the acceptable reproducibility of this method, resulted in its high prognostic value in patients at risk of developing CVD [22]. It has been considered as a reliable outcome measure in clinical trials regarding the effectiveness of lipid-lowering therapy in the reduction of cardiovascular risk assessed with the utilization of cIMT [23]. This assessment is frequently performed in adults diagnosed with RA, who tend to develop atherosclerosis at a younger age than their healthy counterparts and therefore have higher cIMT values [24,25]. Nevertheless, there are conflicting data regarding the utilization of cIMT in the pediatric population. Ilisson et al. [26] found increased cIMT in children at the early stage of JIA, whereas Mani et al. [27] and Ververs et al. [28] did not observe any significant difference between JIA patients and healthy controls. There are no data on cIMT values in healthy children depending on their attitude to the rules of cardiovascular prevention.

The principal objective of this study was to further evaluate the relevance of measuring cIMT in the assessment of cardiovascular risk in JIA patients. Furthermore, the authors attempted to distinguish current lifestyle-related factors that are crucial to be eliminated in order to avoid CVD in this population.

## 2. Materials and Methods

The study group consisted of 45 patients (33 girls and 12 boys) with a median age of 14.0 (IQR 6.0) years, who were diagnosed with JIA in accordance with ILAR criteria [29]. Eighteen of them were classified as oligoarthritis (which is the most common subtype of the disease), nine as enthesitis-related arthritis (ERA) and the remaining eighteen as other subtypes, including systemic-onset JIA as well as RF negative and RF-positive polyarthritis. The results from the study group were compared to 37 age- and sex-matched healthy controls who were recruited from patients hospitalized for non-arthritic reasons (mainly: functional dysfunction of the cardiovascular system, e.g., syncope).

The study database included anthropometric measurements (height, weight and calculated BMI) of all participants. Overweight and obesity were defined as BMI values above the 85th and 95th percentile, respectively. The parents supported by the patients answered a survey regarding the perinatal period, average physical activity (before and during the pandemic), mean screen exposure time (before and during the pandemic), dietary habits (including junk food and soft drinks), exposure to secondhand smoking, and a family history of CVD.

The time from JIA diagnosis to study onset and administered treatment were recorded in the database. However, the possible influence of glucocorticoids was not evaluated due to considerable inconsistencies in the data using both intravenous and systemic steroids in a variable period of time prior to the study onset.

Approximately 3 mL of blood samples were drawn from all patients after 8 to 12 h of fasting. The serum levels of the following parameters were measured: erythrocytes sedimentation rate (ESR), C-reactive protein (CRP), glucose, total cholesterol, high- and low-density lipoprotein, triglycerides, uric acid, and creatinine and alanine transaminase (ALT). The oscillometric measurement of blood pressure was performed several (at least three) times during the hospitalization of each patient. The examination was executed in the sitting position and was preceded by at least 5 min of rest. A sphygmomanometer with an inflatable cuff adjusted to arm length and circumference was utilized for these measurements. Furthermore, every patient received a 12-lead electrocardiogram (speed paper 50 mm per second) with the assessment of heart-rate corrected QT interval using Bazett’s formula. The assessment of patients’ cardiovascular health was supplemented with an echocardiographic evaluation of heart function and structure performed by pediatric cardiologists (JZ, KNJ, JS) using Philips Epiq Elite.

Ultrasound measurements of cIMT were performed using Toshiba Aplio 400 with linear transducer with the frequency 12 MHz by a single trained examiner (MG) who was not blinded to diagnosis. Both right carotid artery (RCA) and left carotid artery (LCA) were assessed and their cIMT values were calculated to percentiles. The procedure was conducted in accordance with Pignoli et al. [30] as described in other Polish studies [31,32]. Each patient was asked to lie in the supine position and rotate their head to the left and right to form a 45° angle between the head and the examined artery. B-mode and Color Doppler function were utilized within the procedure. Each artery was measured three times to calculate the mean cIMT for further evaluation.

Statistical analysis was conducted with the use of Statistica 13.3 software (Statsoft Polska, Kraków, Poland). Spearman’s rank correlation test was performed for variables that were not normally distributed. Group comparisons were made using the Mann–Whitney U test. Relations between categorical variables were assessed using Pearson’s chi-squared test. Values were presented as median with interquartile range (IQR) in brackets. *p* values were adjusted using Tukey HSD test. A valid analysis of potential confounders was not possible due to the sample size. *p* values less than 0.05 were considered significant.

The study was approved by the Bioethics Committee of the Medical University of Lodz, Poland (Approval No. RNN/101/19/KB issued on 12 February 2019). The patients provided their informed consent to join the study. This was obtained in a written (when a patient turned 16 before the participation in the study) or oral form (before 16 years of age).

## 3. Results

The general characteristics of the study group are presented in Table 1. After performing measurements in all 82 patients from the both study and control group, cIMT exceeding the 94th percentile was found in four children with JIA (three boys and one girl). However, the sample size was too small to reach statistical significance for JIA (*p* = 0.06433) as a potential risk factor. Although the duration of arthritis did not affect cIMT values (*p* = 0.74), all patients with increased cIMT were diagnosed long enough to be treated with biologic therapy, specifically adalimumab (*p* = 0.091). Additionally, there was no effect of JIA subtypes on the cIMT results (*p* = 0.489).

The analysis of inflammatory markers shown that 5 out of 18 patients with elevated ESR had cIMT above the 75th percentile. Although it was not yet an absolute abnormality, it did significantly differ from children with ESR within the normal limits, with only 2 out of 27 patients exceeding the 75th percentile (*p* = 0.0673, see Figure 1). Furthermore, increased ESR values correlated with higher RCA percentiles on the edge of statistical significance (r = 0.2922, *p* = 0.051443, see Figure 2). Interestingly, the parallel effect on LCA had weaker strength and relevance (r = 0.2603, *p* = 0.084133). No other remarkable discrepancies in laboratory test results were found within the study. Although the presence of the human leukocyte antigen (HLA) B27 seemed to influence cIMT values (which were abnormal in 3 out of 15 HLA-B27+ patients comparing to 1 out of 20 HLA-B27- individuals), the postulated effect was not statistically significant (*p* = 0.178).

A routine 12-lead electrocardiogram did not reveal any important abnormalities in both study and control groups (no arrhythmia, hypertrophy, ischemia, or long QT syndrome were detected). Similarly, the echocardiographic assessment did not show any cardiac defect or dysfunction. Blood pressure measurements identified one patient with prehypertension in the control group.

Family history of CVD was positive in 20 out of 45 JIA patients compared to 18 out of 37 healthy peers. There was no statistical difference in this parameter between groups.

Exposure to secondhand smoking turned out to be a factor modulating the cardiovascular risk regardless of JIA. Abnormal cIMT was found in 2 out of 11 patients whose parents admitted to smoking in the presence of their children, and it considerably differed (*p* = 0.0278) from the children of non-smoking parents (2 out of 71 patients).

JIA patients appeared to be under- (4/45 vs. 0/37), overweight (11/45 vs. 7/37), or obese (7/45 vs. 1.37) more frequently than their healthy counterparts (*p* = 0.00585). Furthermore, overweight or obese patients were more likely to have no physical activity at all. “Inactive lifestyle” was present in 16 out of 45 JIA patients compared to 7 out 37 healthy peers. Importantly, patients’ parents denied symptoms of JIA as the main factor modulating children’s activity. Nevertheless, the difference did not reach statistical relevance (*p* = 0.0976).

Additionally, an “inactive lifestyle” was also associated with sedentary screen time. Doubling the time of screen exposure during the pandemic markedly resulted (*p* < 0.001) in children not receiving the recommended level of regular physical activity (at least 3 times a week). However, JIA patients with an “inactive lifestyle” had screen time exposure exceeding 3 h a day even before the pandemic (*p* = 0.04403).

## 4. Discussion

Persistent active inflammation appears to increase the risk of developing CVD in adults diagnosed with RA [33]. Can one measure the cardiovascular risk in JIA patients too? The current study presents the positive association between elevated ESR and cIMT values, which was concordant with the number of previous findings [34,35]. However, other studies have denied the discrepancies in cIMT values in adults diagnosed with JIA in their childhood when compared to their healthy counterparts [36]. Del Giudice et al. [37] proposed that exposure to secondhand smoking, a decreased BMI, and elevated homocysteine levels might be considered as potential risk factors of developing abnormal cIMT in children with JIA. Furthermore, Hussain et al. [35] observed a worse lipid profile, left ventricular mass index, and brachial artery flow mediated dilatation. Therefore, they postulated earlier cardiovascular dysfunction in JIA patients. Nonetheless, Breda et al. [38] reported an improvement in cIMT values after a year of treatment of JIA. Hence, the aggressive therapy according to treat-to-target approach [39] might provide an additional benefit of maintaining cardiovascular risk along with the inflammatory activity of JIA.

Singh et al. postulated that ESR may be considered a sensitive marker for the extensiveness and intensity of atherosclerosis in adults [40]. They reported a significant correlation between ESR and cIMT (*p* < 0.0001) and the presence of the atherosclerotic plaque (*p* = 0.026). Children, however, do not frequently present the apparent manifestations of atherosclerosis and its sequelae [41]. The highest cIMT value in the current study (0.53 mm) was detected in an obese patient treated for four years with adalimumab due to therapy-resistant JIA. Importantly, Satija et al. noted a positive correlation (r = 0.432 *p* = 0.015) between ESR and cIMT in JIA patients [42]. Interestingly, the relationship between ESR and cIMT is more evident in adult RA patients with a risk factor for developing CVD (diabetes, hypertension, hypercholesterolemia, obesity, and smoking cigarettes) [43]. On the other hand, Al-Shehhi et al. did not report a significant correlation between cIMT and ESR in Irish patients with RA or psoriatic arthritis [44]. Although the authors did not confirm the effect of lifestyle-related factors on ESR values, the cardiovascular prevention in children with JIA seems to be legitimate.

Despite being more prevalent in boys (3:1), elevated cIMT values were not significantly dependent on gender in this study. Murni et al. [45] indicated the positive association of hyperinsulinemia and hypercholesterolemia with cIMT among boys but it was not observed in girls. Superko et al. [46] noted that chronic inflammation in adults with RA leads to the structural modification of lipoproteins and an unfavorable lipid profile, leading to a so-called atherogenic phenotype, which manifests as decreased high-density lipoprotein, increased triglycerides, and increased small, low-density lipoprotein. Interestingly, cIMT was found to be correlated with low-density lipoprotein in JIA patients by Breda et al. [38]. The authors did not find such a relationship in this study.

On the other hand, smoking and drinking alcohol affected cIMT in adult women but not in men who presented higher cIMT values when diagnosed with hypertension [47]. The laboratory test results from the study initially suggested the potential importance of the presence of HLA-B27, but eventually no significant association was found. Patients with ankylosing spondylitis were reported to have higher cIMT values elsewhere [48], but, similarly, there was no correlation with HLA-B27.

Long QT syndrome may result in the elevated risk of developing cardiac rhythm disturbances and sudden cardiac death in the general population [49]. Voskuyl et al. [4] found RA patients to be at higher risk of these cardiovascular manifestations than healthy controls. However, Koca et al. did not observe a higher incidence of long QT syndrome in JIA patients [6]. Additionally, this study did not reveal any significant abnormalities in the electrocardiogram of JIA patients; however, current state-of-the-art research does not allow for explicit conclusions about cardiac arrhythmias in children with JIA.

Blood pressure is another issue to be addressed in the evaluation of the state of cardiovascular health. The authors did not detect any abnormalities in this matter in JIA patients (only one child from the control group was classified as prehypertension). Breda et al. [38] reported elevated blood pressure in JIA patients, but with the tendency to decrease after a one-year observation. Such an effect may be caused by the better management of underlying JIA within the time interval between these publications. Accordingly, the general health status of JIA patients has been improved within the last few years thanks to better access to treat-to-target therapy [50]. It may also explain why no significant findings regarding blood pressure were found in this study, which did not include children with a new diagnosis but, rather, patients previously diagnosed with JIA with a median duration of 4.0 years.

Can the lifestyle-related factors modulating cardiovascular risk be easily defined? Overall, exposure to secondhand smoking has a deep impact on child development. It is not only considered as a risk factor of developing JIA [51,52] but it essentially increases the probability of CVD developing in adolescents and young adults [53], including those suffering from autoimmune arthritides [54]. Evidence from Young Finns The Cardiovascular Risk revealed that having both parents smoke resulted in vascular age 3.3 years greater in young adulthood when compared to having neither parent smoke [55]. Furthermore, exposure to secondhand smoking was associated with higher cIMT values in the latest meta-analysis [56]. In this study, the effect of passive smoking on cIMT measurements was determined regardless of JIA. The observed influence might have become even stronger during the COVID-19 pandemic, when smokers confessed to having increased their smoking activity, especially during remote working [57]. Moreover, the COVID-19-related restrictions exacerbated the epidemic of overweight and obesity in children and adolescents [58,59]. The excess weight seems to play an important role in developing cardiovascular risk factors in JIA patients, who are more likely to have abnormal BMI [60,61]. However, body composition was not dependent on cIMT in a study of the general pediatric population in Slovenia [62]. The lack of association between cIMT and BMI was also reported by Aggoun et al. [63] and reflected in the results of this study as well.

Nevertheless, regular physical activity may reduce the risk of developing CVD up to 30% [64]. Beukelman et al. reported that the general level of physical activity was lower in JIA patients than in their healthy peers [65]. Children diagnosed with JIA were also observed to have less leisure activity during the week [66]. Due to psychosocial stress triggered by the diagnosis of JIA, a trend towards the “inactive lifestyle” tended to increase over time from disease onset [67]. Therefore, positive health behaviors, including regular physical activity in childhood and adolescence, need to be further promoted [68]. A reduction in sedentary screen time could also potentially have broad health benefits, as it is associated with, e.g., higher energy intake and poor diet quality [69].

Current approaches regarding the personalized treatment of JIA postulates that complex management should be provided by a multidisciplinary team involving trained physicians, nurses, physiotherapists, psychologists, and other allied health professionals [70]. The current article questions whether a pediatric cardiologist should become a regular member of the team in order to manage cardiovascular risks in JIA patients.

## 5. Limitations

The central limitation of this study appears to be its small sample size and its relative heterogeneity. The authors encountered obstacles in recruiting a larger number of patients during the pandemic due to the limitations for non-COVID-19 patients as well as the reluctance of patients’ parents to spend more time in the hospital than necessary. There were also numerous confounders that the authors were unable to measure, which may have influenced the results. The probable effect modifiers include: exposure to glucocorticoids in various ways of administration; age at the onset of treatment; duration of symptoms before diagnosis of JIA; dissimilarities between JIA subtypes; number of older siblings; parents’ marital status; and truthfulness about secondhand smoking exposure or dietary habits. The remaining major restrictions include: selection bias (only consenting patients who were admitted to the department during the recruitment period were included in the study); recall bias (part of the analysis was based on parents’ answers); the subjectivity of ultrasound measurements conducted by a single examiner who was not blinded to diagnosis; and no follow-up assessment after a certain period of time. Future reanalysis would be possible after enlarging the study group and expanding the study team with another physician to cross-check cIMT values.

## 6. Conclusions

Despite the high hopes placed in cIMT as a potential screening marker of higher cardiovascular risk, the results obtained are insufficient to advise readers whether or not to use this method. JIA patients with positive inflammatory activity (including elevated ESR) who are exposed to secondhand smoking might be the group of interest for future research on CVD concurrent with autoimmune arthritides. For now, the promotion of a healthy lifestyle involving regular physical activity (at least 3 times a week) is worth considering in all children and adolescents, including JIA patients.

## Figures and Tables

**Figure 1 children-10-00422-f001:**
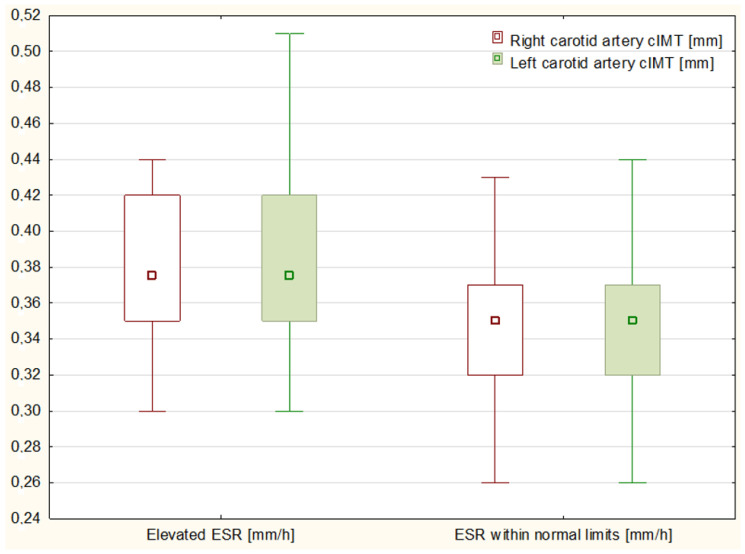
Comparison of carotid intima-media thickness (cIMT) between patients with elevated (**left**) and normal (**right**) erythrocytes sedimentation rate (ESR).

**Figure 2 children-10-00422-f002:**
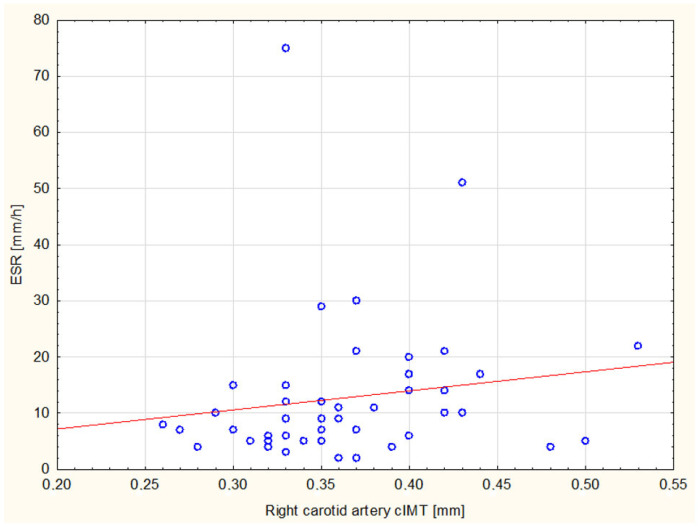
Correlation between erythrocytes sedimentation rate (ESR) values and right carotid artery (RCA) carotid intima-media thickness (cIMT).

**Table 1 children-10-00422-t001:** General characteristics of the study group.

	JIA Patients (*n* = 45)	Patients with cIMT > 94th Percentile (*n* = 4)	Healthy Controls (*n* = 37)
Female:male, *n*	33:12	3:1	23:14
Age, median (IQR)	14.0 (6.0) years	12.5 (6.5) years	14.0 (5.0) years
Duration of JIA, median (IQR)	4.0 (4.0) years	4.5 (5.0) years	-
Abnormal ^1^ BMI, n	23	2	8
JIA subtypes, *n*			
Oligoarticular JIA	18	1	-
RF-negative JIA	14	1	-
RF-positive JIA	2	-	-
Systemic-onset JIA	2	-	-
ERA	9	2	-
Treatment, *n*			
Methotrexate	45	4	-
Other DMARDs	16	2	-
Biological treatment	27	4	-
Laboratory tests results, *n*			
ESR ≥ 20 mm/h	18	2	-
CRP ≥ 5 mg/L	4	-	-
Uric acid ≥ 4 mg/L	20	1	18
Fasting glucose ≥ 100 mg/L	3	-	1
Abnormal ^2^ lipid panel	35	3	28
cIMT measurements			
RCA, median (IQR)	0.35 (0.07) mm	0.49 (0.055) mm	0.38 (0.05) mm
RCA > 75th percentile, *n*	7	4	-
LCA, median (IQR)	0.36 (0.07) mm	0.49 (0.045) mm	0.37 (0.06) mm
LCA >75th percentile, *n*	7	4	-

cIMT—carotid intima-media thickness; JIA—juvenile idiopathic arthritis; IQR—interquartile range; BMI—body mass index; RF—rheumatoid factor; ERA—enthesitis-related arthritis; DMARD—disease-modifying anti-rheumatic drug; ESR—erythrocytes sedimentation rate; CRP—C-reactive protein; RCA—right carotid artery; LCA—left carotid artery. 1—Abnormal BMI encompasses patients with one of the following: underweight (<5th percentile); overweight (>85th percentile); obesity (>95th percentile); 2—Abnormal lipid panel encompasses patients with at least one of the following: total cholesterol ≥ 170 mg/dL; high-density lipoprotein ≤35 mg/dL; low-density lipoprotein ≥ 110 mg/dL; triglycerides ≥ 100 mg/dL.

## Data Availability

The data used to support the findings of this study are included within the article. The data are available from the corresponding author upon request.

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
