# Peer review of "Why and How Should We Assess the Cardiovascular Risk in Patients with Juvenile Idiopathic Arthritis? A Single-Centre Experience with Carotid Intima-Media Measurements"

_children, 2023, doi:10.3390/children10030422_

Round 1
Reviewer 1 Report
Re: Carotid intima-media thickness measurements in children with juvenile idiopathic arthritis – future or fantasy? The assessment of cardiovascular risk factors during the COVID-19 pandemic.
The manuscript reports a study that examined whether carotid intima-media thickness can be used as a novel marker of cardiovascular risk in patients with juvenile idiopathic arthritis. The reported study has serious problems in design, methods, interpretation of results, and reporting. Examples:
Introduction: the confusion starts from the title. The study seems to have two objectives (cIMT association with JIA, and the changes on behavioral CV risk factors due to COVID-19 pandemic). The title implies that the two objectives are related, but they are not. Furthermore, it is not clear how the study addressed the main objective: “to further evaluate the relevance of measuring cIMT in the assessment of cardiovascular risk in JIA patients”—how CV risk was defined and measured is not clear.
Methods: the study suffers from severe flaws in design and analysis. The following are only the major ones:
1- A small sample size, which precludes adequate assessment of the effect of such long-term factors as secondhand smoking and sedentary lifestyle on CV risk, especially given the young age of the study sample. The only excuse is that JIA is a rare disease, but then the authors should acknowledge that several alternative explanations for their findings exist.
2- No comparison was provided on important and relevant characteristics between the cases and the controls.
3- Although Spearman’s correlation and Mann-Whitney U are the appropriate tests to compare not normally distributed variables, no multivariable analysis was performed to account for confounding.
4- Despite the case-control design, not a singe odds ratio (and 95% CI) was reported to measure the magnitude of the effect.
5- With such a small sample size the categorization of cIMT and BMI are not warranted; using them as continuous variables may reveal more information.
6- No adjustment for P for multiple testing was done.
Results:
1- Table 1 does not make much sense because of the very small number of one group, which makes any comparison hard to interpret (proportions in the >94th percentile group are unstable due to the very small numerator). Also, the table is confusing because the 1st column is titled “study group” but both columns are study group. In fact, the term includes the controls, too.
2- The result section is confusing because the authors jump between the results of comparing cases and controls and analyzing only the cases without clearly saying so. For example, it is not clear what the statement “However, the sample size was too small to reach statistical significance for JIA (p=0.06296) or gender (p=0.06637) as potential risk factors” refers to (Page 3, lines 110-112). Because the first result implies a comparison between cases and controls, while at the same time testing the effect of gender is incorrect because it is a matched-on variable.
3- Lumping low and high BMI together is not warranted, especially when the main outcome is cIMT (Page 3, lines 131-134).
4- There is no justification for testing and reporting the association of “inactive lifestyle” with “sedentary screen time” (Page 3, lines 140-141), because it is not relevant to the research questions. Furthermore, the association could be because the two variables measure the same behavior.
5- Although the effect of COVID-19 pandemic is stated as a main objective, only one, vague, pre-post result is mentioned at the very end of the result section, and only for the case group.
Discussion:
1- The discussion needs rewriting to make it shorter, more organized, and based on the findings. For example, the authors reported that all cases with higher cIMT had “biological therapy” (not clear what this means) in “results” section but failed to discuss why this observation is important and what its clinical relevance.
2- Despite all the design limitations and the lack of meaningful findings, the authors still conclude that cIMT measurement “should be considered in all JIA patients” (page 5, line 209). Although cIMT measurement in not invasive, it is still not cheap and takes time and effort, and this recommendation is not supported by the findings from this study.
The manuscript can obviously benefit from involving an epidemiologist/biostatistician in reviewing the study methodology. Also, it needs careful editing for clarity, organization, and word choice, too.
Reviewer 2 Report
I would like to congratulate the authors on their manuscript “Carotid intima-media thickness measurements in children with juvenile idiopathic arthritis – future or fantasy? The assessment of cardiovascular risk factors during the COVID-19 pandemic.”
I read your manuscript with interest but do have some questions.
Title: The title is very long and it confuses me – are you evaluating the Future or Fantasy of Carotid intima-media thickness in JIA or the cardiovascular risk during COVID 19 pandemic? These are two completely different questions and I would recommend using one or the other as your Title.
Abstract:
1) I am unsure how you can say cIMT should be considered in routine CV care for JIA patients. Where is the correlation these measures at this age cause CVD at an older age? For this recommendation you should really follow the patients prospectively and see who develops CVD taking into account all the effect modifiers and confounders. I would strongly recommend to rephrase this statement.
Background
11) I am confused by the reference used to define the risk JIA patients have for CVD. This is not a study showing the risk in these patients but an expert panel. If you want to state there is a risk you have to provide the reference with the proof of this risk – the studies performed in JIA patients showing an increased frequency of CVD compared to health matched controls. If you don’t have a reference like that and this is the only reference you have to show the perceived risk – you should really change the sentence to there is a presumed increased risk as in RA, however this has never been systematically investigated.
22) You quote some studies, however I assume these are cross sectional JIA patients without longitudinal follow-up which means the studies conclude there is an increase cIMT and this is associated in other diseases such as RA to be associated with increased CVD however the association with JIA is still unclear as long-term data is missing…
Materials and methods:
11) I miss the rational why 45 patients? Was that the amount of patients needed to show a difference? Was a power calculation performed. Or was this a sample of convenience? How were patients recruited and were there patients who declined being involved and if so were demographics such as BMI/height/weight different for these patients?
22) Same for the healthy matched controls – where were they consented?
33) There is a perceived bias patients who have a high BMI or weight or poor eating habits will not want to be included and in the methods it is unclear how selection has taken place and if selection biased was avoided as best as possible.
Results
11) I don’t think any statistics should be performed if you only have 4 patients with cIMT findings – the chances you find a correlation just based on chance is too high. – I would strongly suggest not to use p values as potential significant results or not – with only 4 positive findings you should not make any correlations or causalities.
22) Table 1:
a. please provide male and female numbers. Male:female 12:33.
b. Please use median +/- IQR as you can not use mean for a population n=4. This is applicable to all mean values used in this study. Even the n=45 should have lead to the use of median with IQRs and not means. – simple statistic rules for small number populations.
33) I am confused about the smoking – non smoking numbers. 2 out of 17 patients who parents smoked compared to 2 out of 69 patients whose parents did not smoke. 69+17 is 86 and your study is only on 82 patients – where did the additional 4 patients come from?
Discussion:
11) Discussion should be written with more caution as the sample size is so small as are the patients with positive findings in absence of long-term follow-up and no evidence this will lead to CVD in these patients
22) Limitation section should also include the potential bias in this study – selection bias, recall bias and also potential confounders and effect modifiers. Currently only the sample size is noted as a limitation, which is a very limited view of the limitations in this study.
Although very interesting and important, the study should be re-written to clearly represent the very limited conclusions that can be drawn from this study.
Round 2
Reviewer 1 Report
Although few raised points are addressed, most are not. These are serious design problems that have to be addressed (e.g., not adjusting for potential confounders, or not providing a table comparing cases and controls on major health-related characteristics).
Author Response
In our view we have addressed all the Reviewer’s doubts mentioned in the first cycle of Review. We can only apologize for not living up to the Reviewer’s expectations. As we have already discussed, the small sample size do not let us draw strong unbiased conclusions. Of course we are aware of multiple potential confounders and the manuscript has been extended with an extra paragraph as requested.
In the first cycle of review the Referee has evaluated Table 1 as potentially redundant so there was no point in expanding it with supplementary data. After receiving the current comments, we added the proper information on healthy controls to Table 1. We are ready for further improvements if necessary. However, the presented research was a prospective study which has already ended therefore we cannot alter the study design at the recent point.
Reviewer 2 Report
Dear Authors,
Thank you for the revision of your manuscript.
Overall, I agree with the changes made based on the reviewer comments. The major flaw is still the limited amount of patients and no power-calculation or explanation you can draw any conclusions of this sample.
The conclusions drawn are based on a very limited powered study and the language to explain the results, discussion and conclusion needs to be adjusted accordingly. This study is unfortunately not powered to make any recommendations. It just shows a potential but the methodology and patient population recruited does not allow any recommendations.
Author Response
We are very grateful for the Reviewer’s continuous will to help us improve the manuscript. We completely agree with the Reviewer’s assessment of the power of the study. We do believe that prevention of cardiovascular events is better and cheaper than curing them therefore we felt obliged to recommend cIMT screening regardless of the results. However, we agree that our results do not authorize us to do so. The wording has been properly adjusted.